



# Fast domain-aware neural network emulation of a planetary boundary layer parameterization in a numerical weather forecast model

[1]Jiali Wang, [2]Prasanna Balaprakash, and [1]Rao Kotamarthi[*]

[1]Environmental Science Division, Argonne National Laboratory, 9700 South Cass Avenue, Lemont, IL 60439, USA

[2]Mathematics and Computer Science Division, Argonne National Laboratory, 9700 South Cass Avenue, Lemont, IL 60439, USA

*Correspondence to:* Rao Kotamarthi (vrkotamarthi@anl.gov)

**Abstract.** Parameterizations for physical processes in weather and climate models are computationally expensive. We use model output from a set of simulations performed using the Weather Research Forecast (WRF) model to train deep neural networks and evaluate whether trained models can provide an accurate alternative to the physics-based parameterizations. Specifically, we develop an emulator using deep neural networks for a planetary boundary layer (PBL) parameterization in the WRF model. PBL parameterizations are commonly used in atmospheric models to represent the diurnal variation of the formation and collapse of the atmospheric boundary layer—the lowest part of the atmosphere. The dynamics of the atmospheric boundary layer, mixing and turbulence within the boundary layer, velocity, temperature, and humidity profiles are all critical for determining many of the physical processes in the atmosphere. PBL parameterizations are used to represent these processes that are usually unresolved in a typical numerical weather model that operates at horizontal spatial scales in the tens of kilometers. We demonstrate that a domain-aware deep neural network, which takes account of underlying domain structure that are locality specific (e.g., terrain, spatial dependence vertically), can successfully simulate the vertical profiles within the boundary layer of velocities, temperature, and water vapor over the entire diurnal cycle. We then assess the spatial transferability of the domain-aware neural networks by using a trained model from one location to nearby locations. Results show that a single trained model from a location over the midwestern United States produces predictions of wind components, temperature, and water vapor profiles over the entire diurnal cycle and all nearby locations with errors less than a few percent when compared with the WRF simulations used as the training dataset.

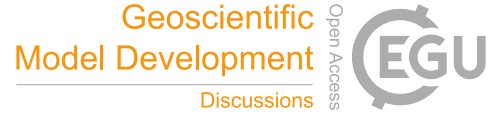

## 1 Introduction

Model developers use approximations to represent the physical processes involved in climate and weather that cannot be resolved at the spatial resolution of the model grids or in cases where the phenomena are not fully understood (Williams, 2005). These approximations are referred to as parameterizations (McFarlane, 2011). While these parameterizations are designed to be computationally efficient, calculation of a model physics package still takes a good portion of the total computational time. For example, in the community atmospheric model (CAM) developed by National Center for Atmospheric Research (NCAR), with spatial resolution of approximately 300 km and 26 vertical levels, the physical parameterizations account for about 70% of the total computational burden (Krasnopolsky and Fox-Rabinovitz, 2006). In the Weather Research Forecast (WRF) model, with spatial resolution of tens of kilometers, time spent by physics is approximately 40% of the computational burden. The input and output overhead is around 20% of the computational time at low node count (100's) and can increase significantly at higher node count as a percentage of the total wall-clock time.

An increasing need in the climate community is performing high spatial resolution simulations (10 km or less grid spacing) and generating large ensembles of these simulations in order to address uncertainty in the model projections and to assess risk and vulnerability. Developing process emulators (Leeds et al., 2013; Lee et al., 2011) that can reduce the time spent in calculating the physical processes will lead to much faster model simulations, enabling researchers to generate high spatial resolution simulations and a large number of ensemble members.

A neural network (NN) is composed of multiple layers of simple computational modules, where each module transforms its inputs to a nonlinear output. Given sufficient data, an appropriate NN can model the underlying nonlinear functional relationship between inputs and outputs with minimal human effort. During the past two decades, NN techniques have found a variety of applications in atmospheric science. For example, Collins and Tossot (2015) developed an artificial NN model by taking numerical weather prediction model (e.g., WRF) output as input to predict thunderstorm occurrence within a few hundreds of square kilometers about 12 hours in advance. Krasnopolsky et al. (2016) used NN techniques for filling the gaps in satellite measurements of ocean color data. Scher (2018) used deep learning to emulate the complete physics and dynamics of a simple general circulation model and indicated a potential capability of



weather forecasts using this NN-based emulator. Neural networks are particularly appealing for
emulations of model physics parameterizations in numerical weather and climate modeling, where
the goal is to find nonlinear functional relationship between inputs and outputs (Cybenko, 1989;
Hornik, 1991; Chen and Chen, 1995a,b; Attali and Pagès, 1997).  NN techniques can be applied
to weather and climate modeling in two ways. One approach involves developing new
parameterizations by using NNs. For example, Chevallier et al. (1998; 2000) developed a new NN-
based longwave radiation parameterization, NeuroFlux, which has been used operationally in the
European Centre for Medium-Range Weather Forecasts four-dimensional variational data
assimilation system and is eight times faster than the previous parameterization. Krasnopolsky et
al. (2013) developed a stochastic convection parameterization based on learning from data
simulated by a cloud-resolving model, CRM, initialized with and forced by the observed
meteorological data. The NN convection parameterization was tested in the NCAR CAM and
produced reasonable and promising results for the tropical Pacific region. Jiang et al. (2018)
developed a deep NN-based algorithm or parameterization to be used in the WRF model to provide
flow-dependent typhoon-induced sea surface temperature cooling. Results based on four typhoon
case studies showed that the algorithm reduced maximum wind intensity error by 60–70%
compared with using the WRF model. The other approach for applying NN to weather and climate
modeling is to emulate existing parameterizations in these models. For example, Krasnopolsky et
al. (2005) developed an NN-based emulator for imitating an existing atmospheric longwave
radiation parameterization for the NCAR CAM. They used output from the CAM simulations with
the original parameterization for the NN training. They found the NN-based emulator was 50–80
times faster than the original parameterization and produced almost identical results.
We study NN models to emulate existing physical parameterizations in atmospheric models.
Process emulators that can reproduce physics parameterization can ultimately lead to the
development of a faster model emulator that can operate at very high spatial resolution as
compared with most current model emulators that have tended to focus on simplified physics
(Kheshigi et al., 1999). Specifically, this study involves the design and development of a domain-
aware NN to emulate a PBL parameterization using 22-year-long output created by a set of WRF
simulations. To the best of our knowledge, we are among the first to apply deep neural networks
to the WRF model to explore the emulation of physics parameterizations. As far as we know from
the literature available at the time of this writing, the only application of NNs for emulating the



parameterizations in the WRF model is by Krasnopolsky et al. (2017). In their study, a three-layer
NN was trained to reproduce the behavior of the Thompson microphysics (Thompson 2008)
scheme in the WRF-ARW model. While we focus on learning the PBL parameterization and
developing domain-aware NN for emulation of PBL, the ultimate goal of our on-going project is
to build an NN-based algorithm to empirically understand the process in the numerical
weather/climate models that could be used to replace the physics parameterizations that were
derived from observational studies. This emulated model would be computationally efficient,
making the generation of large ensemble simulations feasible at very high spatial/temporal
resolutions with limited computational resources. The key objectives of this study are to answer
the following questions specifically for PBL parameterization emulation: (1) What and how much
data do we need to train the model? (2) What type of NN should we apply for the PBL
parameterization studied here? (3) Is the NN emulator accurate compared with the original
physical parameterization? This paper is organized as follows. Section 2 describes the data and the
neural network developed in this study. The efficacy of the neural network is investigated in
Section 3. Discussion and summary follow in Section 4.

## 108   2  Data and Method

### 109   2.1 Data

The data we use in this study is output from the regional climate model WRF version 3.3.1. WRF
is a fully compressible, nonhydrostatic, regional numerical prediction system with proven
suitability for a broad range of applications. The WRF model configuration and evaluations are
given by Wang and Kotamarthi (2014). Covering all the troposphere are 38 vertical layers, between
the surface to approximately 16 km (100 hPa). The lowest 17 layers cover from the surface to
about 2 km above the ground. The PBL parameterization we used for this WRF simulation is
known as the YSU scheme (Yonsei University; Hong et al., 2006). The YSU scheme uses a
nonlocal-mixing scheme with an explicit treatment of entrainment at the top of the boundary layer
and a first-order closure for the Reynolds-averaged turbulence equations of momentum of air
within the PBL.
We use the output of the WRF model driven by NCEP-R2 for the period 1984–2005. The 22-year
data was partitioned into three parts: a training set consisting of 20 years (1984–2003) of 3-hourly



data to train the NN; a development set (also called validation set) consisting of 1 year (2004) of
3-hourly data used to tune the algorithm's hyperparameters and to control overfitting (the situation
where the trained network predicts well on the training data but not on the test data); and a test set
consisting of 1 year of records (2005) for prediction and evaluations. The goal of the work
described here is to develop an NN-based parameterization that can be used to replace the PBL
parameterization in the WRF model. Thus, we expect the NN submodel to receive a set of inputs
that are equivalent to the inputs provided to the YSU scheme at each timestep. However, a key
difference in our approach is that the vertical profiles of various state variables are reconstructed
by the NN using only the inputs (near-surface variables and 700 hPa geostrophic winds).
Table 1 shows the architecture in terms of inputs and outputs used in our experiments. The inputs
are near-surface characteristics including 2-meter water vapor, 2-meter air temperature, 10-meter
zonal and meridional wind, ground heat flux, incoming shortwave radiation, incoming longwave
radiation, PBL height, sensible heat flux, latent heat flux, surface friction velocity, ground temp,
soil temperature at 2 m below the ground, soil moisture at 0–0.3cm below the ground, and
geostrophic wind component at 700 hPa. The outputs for the NN architecture are the vertical
profiles of the following model prognostic and diagnostic fields: temperature, water vapor mixing
ratio, and zonal and meridional wind (including speed and direction). In this study we develop an
NN emulation of the PBL parameterization; hence we focus only on predicting the profiles within
the PBL, which is on average around 200 m and 400 m during the night and afternoon of winter,
respectively, and around 400 m and 1300 m during the night and afternoon of summer,
respectively, for the locations studied here. The middle and upper troposphere (all layers above
the PBL) are considered fully resolved by the dynamics simulated by the model and hence not
parameterized. Therefore, we do not consider the levels above PBL height because (1) they carry
no information about input/output functional dependence that affects the PBL and (2) if not
removed, they introduce additional noise in training. Specifically, we use the WRF output from
the first 17 layers, which are within 1,900 meters and well cover the PBL.

## 2.2 Deep neural networks for PBL parameterization emulation

A class of machine learning approaches that is particularly suitable for emulation of PBL
parameterization is supervised learning. This approach models the relationship between the
outputs and independent input variables by using training data $(x_i, y_i)$, for $x_i \in T \subset D$, where T is



a set of training points, D is the full data set, and $x_i$ and $y_i = f(x_i)$ are inputs and its corresponding
output $y_i$, respectively. The function $f$ that maps the inputs to the outputs is typically unknown and
hard to derive analytically. The goal of the supervised learning approach is to find a surrogate
function $h$ for $f$ such that the difference between $f(x_i)$ and $h(x_i)$ is minimal for all $x_i \in T$. Many
supervised learning algorithms exist in the machine learning literature. In this paper, we focus on
deep neural networks (DNNs).
DNNs are composed of neural layers: a stack of nodes organized in a hierarchical way to model a
nonlinear function. Within each neural layer, nodes receive inputs from previous neural layers and
perform certain nonlinear transformations through a system of weighted connections on the
received input values. The training data is given to the neural network through the input neural
layer. The last neural layer of the stack in the network is the output neural layer from which the
predicted values are obtained. The training procedure consists of modifying the weights of the
connections in the network to minimize a user-defined objective function that measures the
prediction error of the network. Each iteration of the training procedure comprises two phases: the
forward pass consists of passing the training data to the network and computing the prediction
error; in the backward pass, the gradients of the error function with respect to all the weights in
the network is computed and used to update the weights in order to minimize the error. Once the
entire dataset is passed both forward and backward through the neural network (with many
iterations), one epoch is completed.
We consider three variants of DNN (see below). We construct all of them using a neural block that
comprises a dense neural layer with $N$ nodes and a rectified linear activation function, where $N$ is
user-defined parameters.
**Naïve DNN:**
**Deep feed-forward neural network (FFN):** This is a fully connected feed-forward deep neural
network constructed as a sequence of $K$ neural blocks, where the input of the $i$th neural block is
from $\{i\text{-}1\}$th block and the output of the $i$th neural block is given as the input of the $\{i\text{+}1\}$th neural
block. The sizes of the input and output neural layers are 16 (= near-surface variables) and 85 (=
17 vertical levels $\times$ 5 output variables). See Figure 1a for an illustration.
**Domain-aware DNN:**





While the FFN is a typical way of applying NN for finding the nonlinear relationship between
input and output, a key drawback of the naïve FFN is that it does not consider the underlying PBL
domain structure, such as the patterns that are locality specific and the vertical dependence between
different vertical levels of each profile. This is not typically needed for NNs in general and in fact
is usually avoided because, for classification and regression, one can find visual features regardless
of their locations. For example, a picture can be classified as a certain object even that object has
never appeared in the given location in the training set. In our case, however, the location is fixed
and the profiles over that location is distinguishable from other locations if they have different
terrain conditions. Consequently, we want to learn the particular influence of location in the
forecast. For example, the feature at a lower level of a profile plays a role in the feature at a higher
level and can help refine the output at the higher level and accordingly the entire profile. This
dependence may inform the NN and provide better accuracy and data efficiency. To that end, we
develop two variants of domain-aware DNNs for PBL emulation.
**Hierarchically connected network with previous layer only connection (HPC):** We assume
that the outputs at each altitude level depend not only on the 16 near-surface variables but also on
the altitude level below it. To model this explicitly, we develop a domain-aware DNN variant in
which 17 neural blocks are connected as follows: the input to an $i$th ($i>1$) neural block comprises
the input neural layer of the 16 near-surface variables and the 5 outputs of the $\{i\text{-}1\}$th neural block.
The first neural block, which is next to the input layer, receives inputs only from the input neural
layer of the 16 near-surface variables. See Figure 1b for an example.
**Hierarchically connected network with all previous layers connection (HAC):** We assume that
the outputs at each PBL depend not only on the 16 near-surface variables but also on all altitude
levels below it. The input to an $i$th neural block comprises the input neural layer of the 16 near-
surface variables and all outputs of the $\{1, 2, …, i\text{-}1\}$ neural blocks below it. See Figure 1c for an
example.
**2.3 Setup**
For preprocessing, we applied StandardScaler (removes the mean and scales each variable to unit
variance) and MinMaxScaler (scales each variable between 0 and 1) transformations before





training, and we applied the inverse transformation after prediction so that the evaluation metrics
are computed on the original scale.
We note that there is no default value for $N$ nodes in a dense neural layer. We conducted an
experimental study on FFN and found that setting $N$ to 16 results in good predictions. Therefore,
we used the same value of $N = 16$ in HPC and HAC.
For the implementation of DNN, we used Keras (version 2.0.8), a high-level neural network
Python library that runs on the top of the TensorFlow library (version 1.3.0). We used the scikit-
learn library (version 0.19.0) for the preprocessing module. The experiments were run on a Python
(Intel distribution, version 3.6.3) environment.
All three DNNs used the following setup for training: optimizer = adam, learning rate = 0.001,
epochs = 1000, batch size = 64. Note that batch size and number of epochs define the number of
randomly sampled training points required before updating the model parameters and the number
times that training will work through the entire training dataset. To avoid overfitting issues in
DNNs, we use an early stopping criterion in which the training stops when the validation error
does not reduce for 10 subsequent epochs.
We ran training and inference on a NVIDIA DGX-1 platform: Dual 20-Core Intel Xeon E5-2698
v4 2.2 GHz processor with 8 NVIDIA P100 GPUs with 512 GB of memory. The DNN's training
and inference leveraged only a single GPU.

## 3 Results

In the following discussion we evaluate the efficacy of the three DNNs by comparing their
prediction results with WRF model simulations. We refer to the results of WRF model simulations
as observations because the DNN learns all the knowledge from the WRF model output, not from
in situ measurements. We refer to the values from the DNN models as predictions. We initiate our
DNN development at one grid cell from WRF output that is close to a site in the midwestern United
States (Logan, Kansas, latitude= 38.8701°N; longitude= 100.9627°W) and another grid cell at a
site in Alaska (Kenai Peninsula Borough, AK, latitude= 60.7237 °N; longitude=150.4484 °W) to
evaluate the robustness of the developed DNNs. We then apply our DNNs to stations within 800
km from the Logan site to assess the spatial transferability of the DNNs. While the Alaska site has



different vertical profiles, especially for wind directions, and lower PBL heights in both January
and July, the conclusion in terms of the model performance is similar to the site over Logan,
Kansas.

## 3.1 DNN performance in temperature and water vapor

Figure 2 shows the diurnal variation (explicitly 3 PM and 12 AM local time at Logan, Kansas) of
temperature and water vapor mixing ratio vertical profiles in the first 17 layers from the
observation and three DNN model predictions. The 17 layers are within 1,900 meters and well
cover the PBL. The figures present results for both January and July. The dashed lines show the
lowest and highest (5th and 95th percentile, respectively) PBL heights for that particular time. In
general, the DNNs are able to produce similar shapes of the observed profiles, especially within
the PBL. Both the temperature and water vapor mixing ratio are lower in January and higher in
July. Within the PBL, the temperature and water vapor do not change much with height; above the
PBL to the entrainment zone, the temperature and water vapor start decreasing. Among the three
DNNs, HAC and HPC show very low bias and high accuracy in the PBL, but the FFN shows a
relatively large discrepancy from the observation. Figure 3 shows the root-mean-square error
(RMSE) and Pearson correlation coefficient (COR) between observation and three DNN
predictions in the afternoon and midnight of January and July. The RMSE and COR consider not
only the time series of observation and prediction but also their vertical profiles below the PBL
heights for each particular time. Among the three DNNs, HPC and HAC always show better skill
with smaller RMSEs and higher CORs than does FFN. The reason is that the FFN uses only the
16 near-surface variables as inputs and does not consider dependence between each of the vertical
levels. In contrast, HPC and HAC use not only the near-surface variables but also the five variables
of one previous vertical level (HPC) or all previous vertical levels (HAC) as inputs for predicting
the profiles of each field. This approach is important because PBL parameterizations are used to
represent the vertical dependence of these variables and are usually unresolved in a typical climate
and weather models that operate at horizontal spatial scales in the tens of kilometers. Compared
with HAC, HPC sometimes shows slightly better accuracy with smaller RMSEs and higher CORs,
but in other cases HPC performs similar to HAC. These results indicate that the information from
all previous levels is not as important as information from the previous layer right below the
predicted layer.

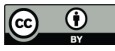



## 3.2 DNN performance in wind component

Figure 4 shows the diurnal variation of wind (including wind speed and direction) profiles in January and July 2005 from observation and three DNN predictions. Compared with the temperature and water vapor profiles, the wind profiles are more difficult to predict, especially for days (e.g., summer) that have a higher PBL. The wind direction does not change much below the majority of the PBL, and it turns to westerly winds when going up and beyond the PBL. The DNN prediction has difficulty predicting the profile above the PBL height, as is expected because these layers are considered fully resolved by the dynamics simulated by the WRF model and hence not parameterized. Therefore, we do not consider DNN performance at the levels above PBL height, because the DNNs carry no information about input/output functional dependence that affects the PBL. The wind speed increases with height in both January and July within the PBL. Above the PBL heights, the wind speed still increases in January but decreases in July. The reason is that in January the zonal wind, especially westerly wind, is dominant in the atmosphere and the wind speed increases with height; in July, however, the zonal wind is relatively weak, and the meridional wind is dominant with southerly wind below ~2 km and northerly wind above 2 km. The decrease in wind speed above the PBL is just about the transition of wind direction from southerly to northerly wind. Figure 5 shows the RMSEs and CORs between the observed and predicted wind component within the PBL. The wind component is fairly well predicted especially by the HAC and HPC networks with correlation above 0.8 for wind speed and 0.7 for wind direction except in July at midnight, which is below zero. Similar to the predictions for temperature and water vapor, the FFN shows the poorest prediction accuracy, especially for wind direction. For accurately predicting the wind direction, we found that using the geostrophic wind at 700 hPa as one of the inputs for the DNNs is important.

## 3.3 DNN dependence on length of training period

Next, we evaluate how sensitive the DNN is to the amount of available training data and how much data one would need in order to train a DNN. While we present Figures 2–5 using 20-year (1984–2003) training data, here we gradually decrease the length of the training set to 12 (1992–2003), 6 (1998–2003), 2 (2002–2003) years, and 1 (2003) year. The validation data (for tuning hyper-parameters and controlling overfit) and the test data (for prediction) are kept the same as in our standard training dataset, which is year 2004 and 2005, respectively. Figures 6 and 7 show the

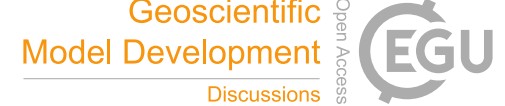

RMSE and CORs between observed and predicted profiles of temperature, water vapor, and wind
component for January and July at their local midnight. Overall, the FFN network depends heavily
on the length of training dataset. For example, the RMSE of FFN predicted temperature decreases
from 7.2 K using one year of training data to 3.0 K using 20-year training data. HAC and HPC
also depend on the length of training data especially when less than 6-year training data is
available, but even their worst prediction accuracy (using one year of training data) is still better
than FFN using 20-year training data. The RMSEs of HPC and HAC predicted temperature
decrease from ~2.4 using 1 year of training data to ~1.5 using 20 years of training data. The CORs
of FFN predicted temperature increase from 0.73 using one year of training data to 0.92 using 20
years of training data. The CORs for HPC and HAC increase slightly with more training data, but
overall they are above 0.85 using one year to 20 years of training data.
Regarding the question about how much data one would need to train a DDN, for FFN, at least
from this study, the performance is not stable until one has 12 or more years of training data, which
is significantly better than having only 6 years or less of training data. For HAC and HPC, however,
having 6 years of training data seems sufficient to show a stable prediction. Increasing the amount
of training data shows only marginal improvement in predictive accuracy. In fact, in contrast to
HAC and HPC, the performance of FFN has not reached a plateau even with the 20 years of
training data. This suggests that with longer training sets the predicting skill of an even naïve
approach like FFN could be further improved and eventually reach the accuracy of HAC and HPC
using 6 or more years of training data.
**3.4 DNN performance for nearby stations**
This section assesses the spatial transferability of the domain-aware neural networks (specifically
HAC and HPC) by using a trained model from one location (at Logan, Kansas, as presented above)
to other locations within 800 kilometers from the Logan site with different terrain conditions and
vegetation types. We choose ten locations, as shown in Figure 8, among which two (Sites 1 and 2)
are 300 km away from the Logan site; three (Sites 3, 4, and 5) are 430 km away from the Logan
site; and five (Sites 6 to 10) are 450–800 km away from the Logan site, with Sites 9 and 10 the
furthest and having the most different elevations from the Logan site. Different from the preceding
section, here we calculate normalized RMSEs relative to each site's observations at a particular
time, in order to make the comparison feasible between different sites. As shown in Figures 9 and





10 by the normalized RMSEs and Pearson correlations, in general, when going farther from Logan
site, where our domain-aware neural networks (HPC and HAC) were developed, the prediction
skill either does not change or gets slightly worse depending on the locations and the difference in
terrain conditions between the reference site (Logan, Kansas) and the remote sites (S1 to S10 in
Figure 8). For example, the RMSEs for wind direction over Sites 2, 4, and 8 are similar to that
over the Logan site. However, the RMSEs over the other sites, which have different elevations
(either higher or lower) than that for Logan site, are much larger, suggesting the DNNs developed
based on Logan site are not applicable for these locations. These results indicate that, at least for
this study, as long as the terrain conditions (slope, elevation, and orientation) are similar, the DNNs
can be applied with similar prediction skill for locations that are as far as 520 km (equal to more
than 40 grid cells in the WRF output used in this study) to predict the wind and also other variables
assessed in this study. The results also suggest that when implementing the NN-based algorithm
into the WRF model, if a number of grid cells are over a homogenous region, one may not need to
train the NN over every grid cell. This will significantly save computing time because the training
process takes the majority of the computing resource (see below). Similar to Figure 6, we see that
the HPC network works better than HAC especially for temperature and water vapor over all the
sites and for wind component over most of the sites examined here, indicating that the input from
all previous layers is not as important as that from the input from only the layer next to the predicted
layer.

## 3.5 DNN training and prediction time

Table 2 shows the number of epochs and time required for training FNN, HPC, and HAC for
various numbers of training years. Because of the early stopping criterion, the number of training
epochs performed by different methods is not same. Despite setting the maximum epochs to 1,000,
all these methods terminate within 178 epochs. We observed that HPC performs more training
epochs than do FFN and HAC: given the same optimizer and learning rate for all the methods,
HPC has a better learning capability because it can improve the validation error more than HAC
and FNN can. For a given set of training data, the difference in the training time per epoch can be
attributed to the number of trainable parameters in FNN, HPC, and HAC (10,693, 16,597, and
26,197, respectively). As we increase the size of training data, the training time per epoch increases
significantly for all three DNN models. The increase also depends on the number of parameters in



the model. For example, increasing the training data from 1 year to 20 years increases the training
time per epoch from 1.4 seconds to 11.4 seconds for FNN, from 1.1 seconds to 17.4 seconds, and
from 1.4 seconds to 19.6 seconds for HPC and HAC, respectively.
The prediction times of FNN, HPC, and HAC are within 0.5 seconds for one-year data, making
these models promising for PBL emulation deployment. The difference in the prediction time
between models can be attributed to the number of parameters in the DNNs: the larger the number
of parameters, the higher the prediction time. For example, the prediction times for FFN are below
0.2 seconds when using different numbers of years for training, while those for HAC are around
0.4 seconds. Despite the difference in the number of training years, the number of parameters for
a given model is fixed. Therefore, once the model is trained, the DNN prediction time depends
only on the model and the number of points in the test data (1 year in this study). Theoretically,
for the given model and the test data, the prediction time should be constant even with different
amounts of training dataset. However, we observed slight variations in the prediction times that
range from 0.17 to 0.29 seconds for FNN, 0.30 to 0.34 seconds for HPC, and 0.36 to 0.42 seconds
for HAC, which can be attributed to the system software.

## 4    Summary and Discussion

This study developed DNNs for emulating a PBL parameterization that is used by the WRF model.
Two of the DDNs take into account the domain-specific features such as spatial dependence in the
vertical direction over the location where we develop the DNNs. The input and output data for the
DNNs are taken from a set of 22-year-long WRF simulations. We developed the DNNs based on
a midwestern location in the United States. We found that the domain-aware DNNs can reproduce
the vertical profiles of wind, temperature, and water vapor mixing ratio with high accuracy yet
require fewer data than the traditional DNN, which does not care about the domain-specific
features. The training process takes the majority of the computing time. Once trained, the model
can quickly predict the variables with decent accuracy. This ability makes the deep neural network
appealing for parameterization emulator.
Following the same architecture that we applied for Logan, Kansas, we also built DNNs for one
location at Alaska. The results share the same conclusion as we have seen for the Logan site. For
example, among the three DNNs, HPC and HAC show much better skill with smaller RMSEs and

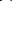



higher correlations than does FFN. The wind profiles are more difficult to predict than the profiles
of temperature and water vapor. For FFN, the prediction accuracy increases with more training
data; for HPC and HAC, the prediction skill stays similar when having six or more years of training
data.
While we trained our DNNs over individual locations in this study using only one computing node
(with multiple processors), there are 300,000 grid cells over our WRF model domain, which
simulated the North American continent as a horizontal resolution of 12 km. To train a model for
all the grid cells or all the homogeneous regions over this large domain, we will need to scale up
the algorithm to hundreds if not thousands of computing nodes to accelerate the training time and
the make the entire NN-based simulation faster than the original parameterization.
The ultimate goal of this project is to build an NN-based algorithm to empirically understand the
process in the numerical weather and climate models and to replace the PBL parameterization and
other time-consuming parameterizations that were derived from observational studies. This
emulated model thus would be computationally efficient and enable researchers to generate large
ensemble simulations at very high spatial/temporal resolutions with limited computational
resources. The DNNs developed in this study can provide numerically efficient solutions to a wide
range of problems in environmental numerical models where lengthy, complicated calculations
describing physical processes must be repeated frequently or need a large ensemble of simulations
to represent uncertainty. A possible future direction for this research is implementing these NN-
based schemes in WRF for a new generation of hybrid regional-scale weather/climate models that
fully represent the physics at a very high spatial resolution at a fast computational time so as to
provide the means for generating large ensemble model runs.
*Data and code availability*. The data used and the code developed in this study are available at
https://github.com/pbalapra/dl-pbl.
*Author contributions*. JW participated in the entire project by providing domain expertise and
analyzing the results from the NN-based emulator. PB developed the deep neural networks and
did all the experiments presented in this study. RK proposed the idea of this project and provided
high-level guidance and insight for the entire study.





*Competing interests*. The authors declare that they have no conflict of interest.

*Acknowledgments*. The WRF model output was developed through computational support by the
Argonne National Laboratory Computing Resource Center and Argonne Leadership Computing
Facility. This material is based upon work supported by the U.S. Department of Energy, Office
of Science, under contract DE-AC02-06CH11357.

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

## 508    **Figure captions**

**Figure 1: Three variants of DNN developed in this study: (a) fully connected feed forward**
**neural network (FFN), (b) hierarchically connected network with previous layer only**
**connection (HPC), and (c) hierarchically connected network with all previous layers**
**connection (HAC).**
**Figure 2: Temperature and water vapor mixing ratio from the observation and three DNN**
**predictions: FFN, HPC, and HAC in January and July 2005 at 3 PM and 12 AM local time.**
**The y-axis uses log scale. The training data are from 20 years (1984 to 2003) of 3-hourly WRF**
**output. The lower and upper dash lines show the lowest and highest (5th and 95th percentile)**
**PBL heights at that particular time. For example, in January 12 AM, the lowest PBL height**
**is about 19 m, while the highest PBL height is about 365 m.**
**Figure 3: RMSE and correlations for time series of temperature and water vapor vertical**
**profiles within the PBL predicted by the three DNNs compared with the observations. The**
**vertical lines show the range of RMSEs and correlations when considering the lowest and**
**highest PBL heights at each particular time (shown by the dashed horizontal lines in Figure**
**2). The training data are 3-hourly WRF output from 1984 to 2003.**
**Figure 4: Same as Figure 2 but for wind direction and wind speed.**
**Figure 5: Same as Figure 3 but for wind components.**
**Figure 6: RMSEs for temperature, water vapor, and wind components at midnight of**
**January using three DNNs. Left y-axis is for RMSEs of HAC and HPC; right y-axis is for**
**RMSE of FFN. The RMSEs are calculated along the time series below the PBL height for**
**January midnight at local time. The lower and upper end of the dash lines are RMSEs that**
**consider the lowest and highest PBL heights as shown in Figure 2.**





**Figure 7: Same as Figure 6 but for Pearson correlations.**

**Figure 8: Terrain height (left) and vegetation types (right) for Logan, Kansas, and other locations that we used to assess the spatial transferability of our domain-aware DNNs.**

**Figure 9: Normalized RMSEs relative to their corresponding observations at midnight of January for temperature, water vapor mixing ratio, and wind component. The sites are in the order of short to long distance from the reference site at Logan, Kansas.**

**Figure 10: Same as Figure 9 but for correlations between DNN predictions and observations.**





**Table 1: Inputs and outputs for the NN developed in this study. The variable names of these**
**inputs and outputs in the WRF are shown in the parentheses.**

| Input Variable | Output Variable |
| --- | --- |
| 2-meter water vapor mixing ratio (Q2), | zonal wind (U) |
| 2-meter air temperature (T2), | meridional wind (V) |
| 10-meter zonal and meridional wind (U10, V10) | temperature (tk) |
| Ground heat flux (GRDFLX) | water vapor mixing ratio (QVAPOR) |
| Downward short wave flux (SWDOWN) | |
| Downward long wave flux (GLW) | |
| Latent heat flux (LH) | |
| Upward heat flux (HFX) | |
| Planetary boundary layer height (PBLH) | |
| Surface friction velocity (UST) | |
| Ground temp (TSK) | |
| Soil temperature at 2 m below ground (TSLB) | |
| Soil moisture for 0-0.3cm below ground (SMOIS) | |
| Geostrophic wind component at 700 hPa (Ug, Vg) | |






**Table 2: Training and prediction time (unit: seconds) for the three DNNs using different**
**lengths of training data. The predicted period is for one year (2005).**

| DNN Type | Training Data (years) | Training Time (s) | Number of Epochs | Training Time (s) per Epoch | Prediction Time (s) for 1 Year (2005) |
|---|---|---|---|---|---|
| FNN | 1 | 85.969 | 61 | 1.409 | 0.197 |
| FNN | 2 | 137.359 | 47 | 2.923 | 0.196 |
| FNN | 6 | 376.209 | 70 | 5.374 | 0.171 |
| FNN | 12 | 199.468 | 23 | 8.673 | 0.193 |
| FNN | 20 | 306.665 | 27 | 11.358 | 0.199 |
| | | | | | |
| HPC | 1 | 199.152 | 178 | 1.119 | 0.336 |
| HPC | 2 | 454.225 | 91 | 4.991 | 0.343 |
| HPC | 6 | 1233.908 | 133 | 9.278 | 0.317 |
| HPC | 12 | 1225.880 | 88 | 13.930 | 0.302 |
| HPC | 20 | 1181.716 | 68 | 17.378 | 0.331 |
| | | | | | |
| HAC | 1 | 131.104 | 95 | 1.380 | 0.366 |
| HAC | 2 | 468.884 | 85 | 5.516 | 0.411 |
| HAC | 6 | 870.753 | 80 | 10.884 | 0.406 |
| HAC | 12 | 737.921 | 47 | 15.700 | 0.420 |
| HAC | 20 | 1351.898 | 69 | 19.593 | 0.381 |






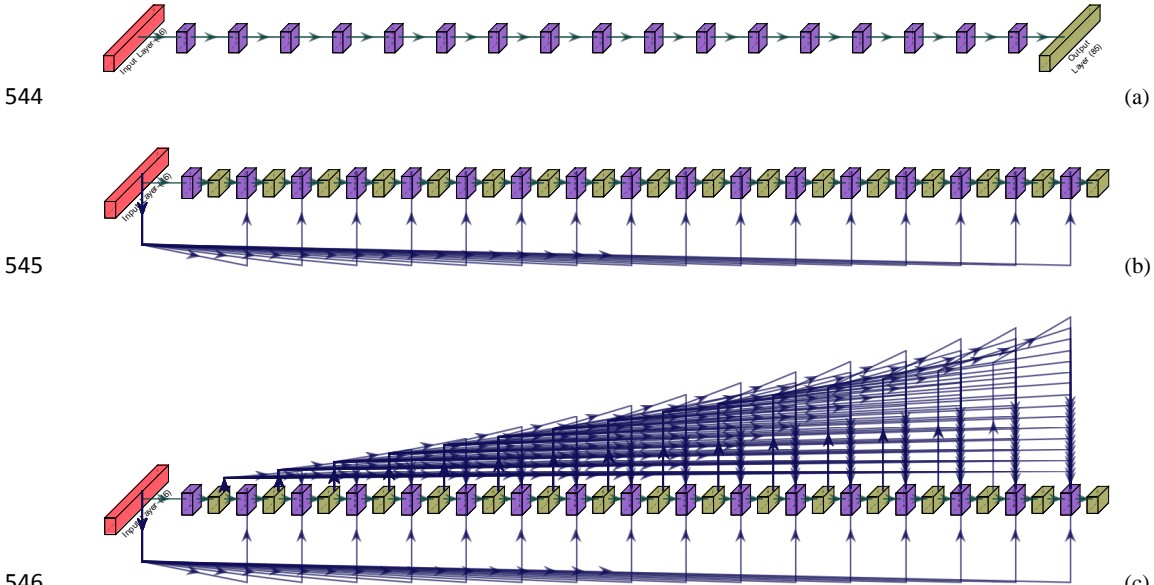

544                                              (a)

545                                              (b)

546                                              (c)

**Figure 1: Three variants of DNN developed in this study: (a) fully connected feed forward**
**neural network (FFN), (b) hierarchically connected network with previous layer only**
**connection (HPC), and (c) hierarchically connected network with all previous layers**
**connection (HAC).**





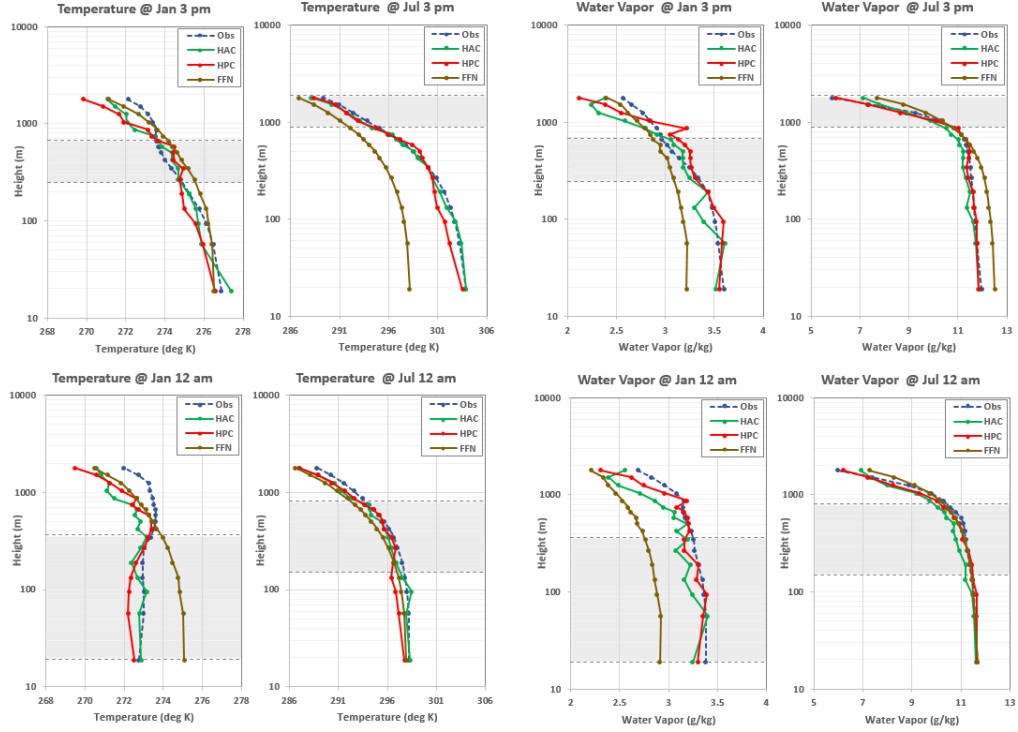


**Figure 2: Temperature and water vapor mixing ratio from the observation and three DNN predictions: FFN, HPC, and HAC in January and July of 2005 at 3 PM and 12 AM local time. The y-axis uses log scale. The training data are from 20 years (1984 to 2003) of 3-hourly WRF output. The lower and upper dash lines show the lowest and highest (5th and 95th percentile) PBL heights at that particular time. For example, in January 12 AM, the lowest PBL height is about 19 m, while the highest PBL height is about 365 m.**





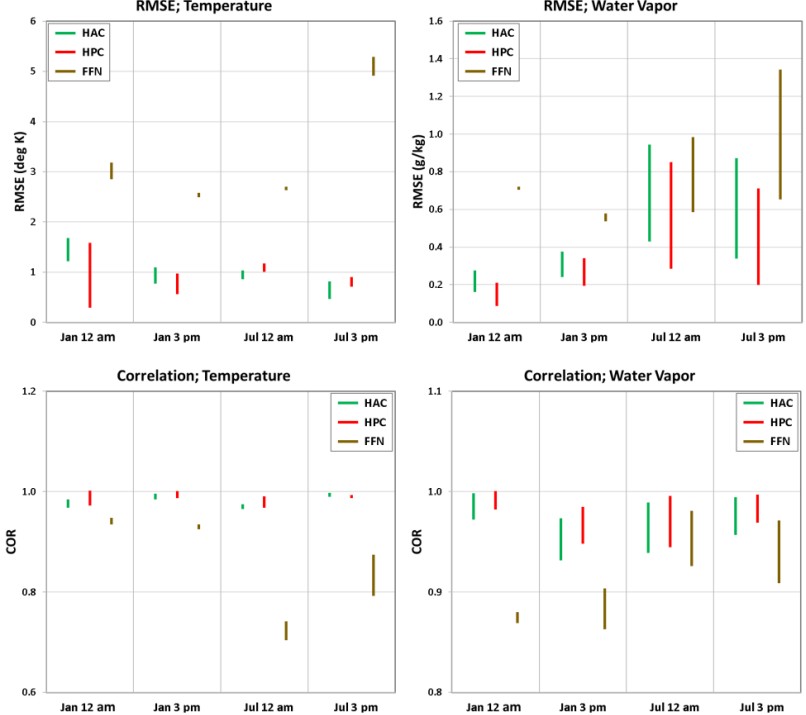


**Figure 3: RMSE and correlations for time series of temperature and water vapor vertical profiles within the PBL predicted by the three DNNs compared with the observations. The vertical lines show the range of RMSEs and correlations when considering the lowest and highest PBL heights at each particular time (shown by the dashed horizontal lines in Figure 2). The training data are 3-hourly WRF output from 1984 to 2003.**






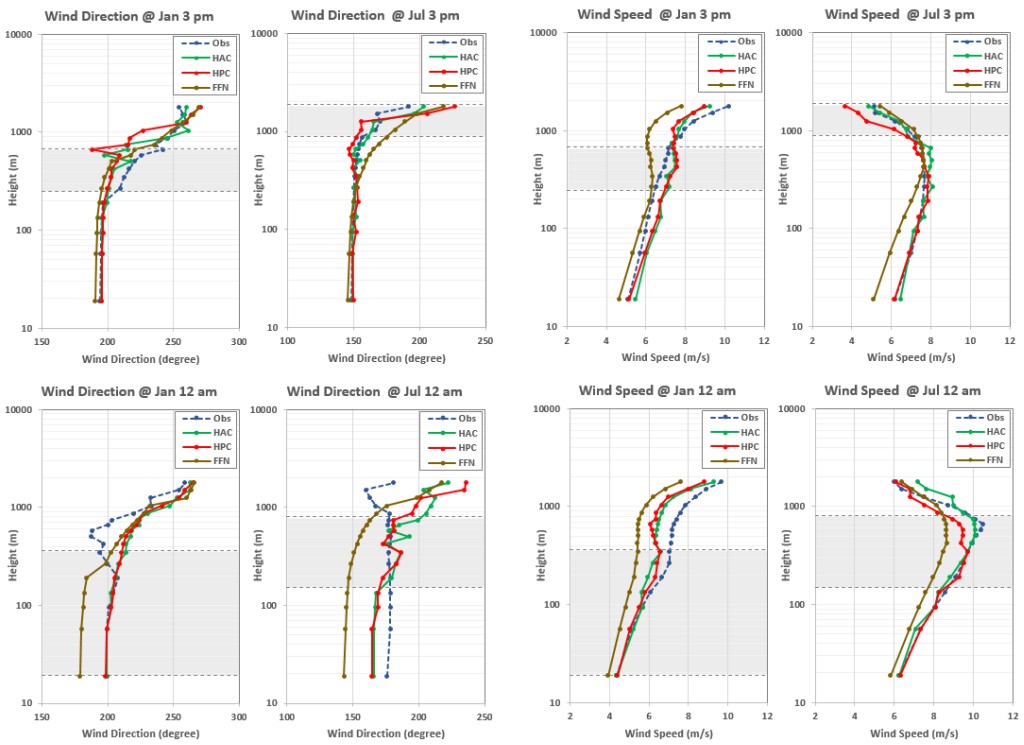


**Figure 4: Same as Figure 2 but for wind direction and wind speed.**



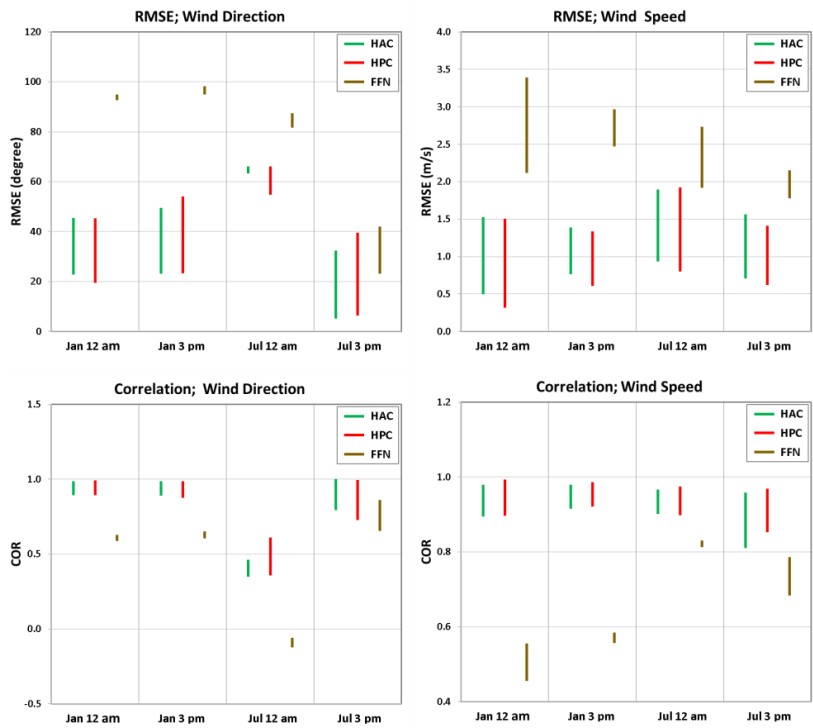


**Figure 5: Same as Figure 3 but for wind components.**





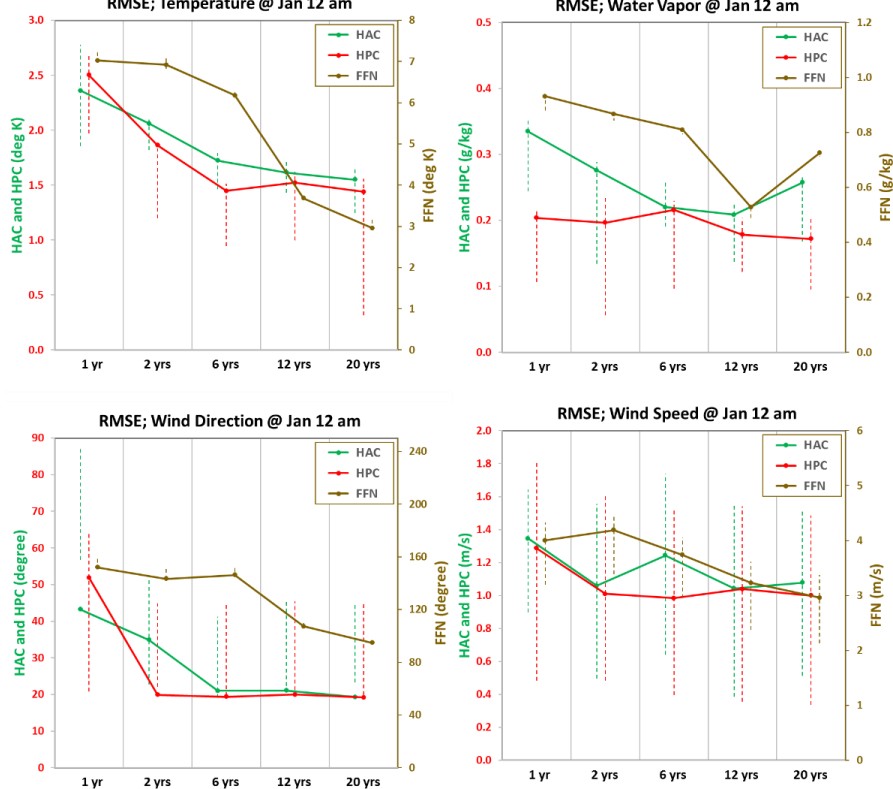


**Figure 6: RMSEs for temperature, water vapor, and wind components at midnight of January using three DNNs. Left y-axis is for RMSEs of HAC and HPC; right y-axis is for RMSE of FFN. The RMSEs are calculated along the time series below the PBL height for January midnight at local time. The lower and upper end of the dash lines are RMSEs that consider the lowest and highest PBL heights as shown in Figure 2.**






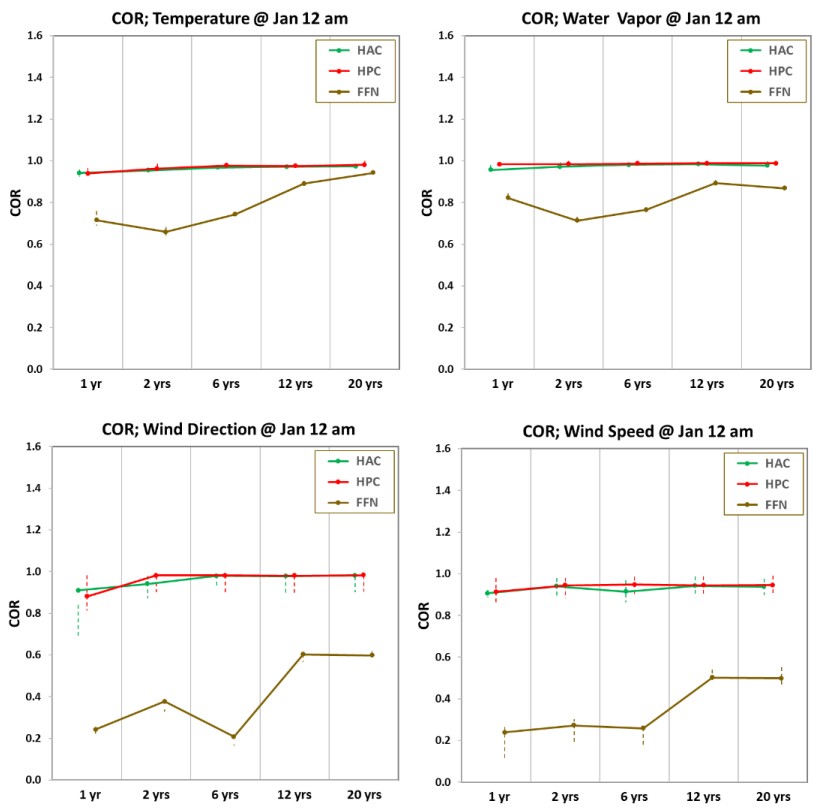


**Figure 7: Same as Figure 6 but for Pearson correlations.**





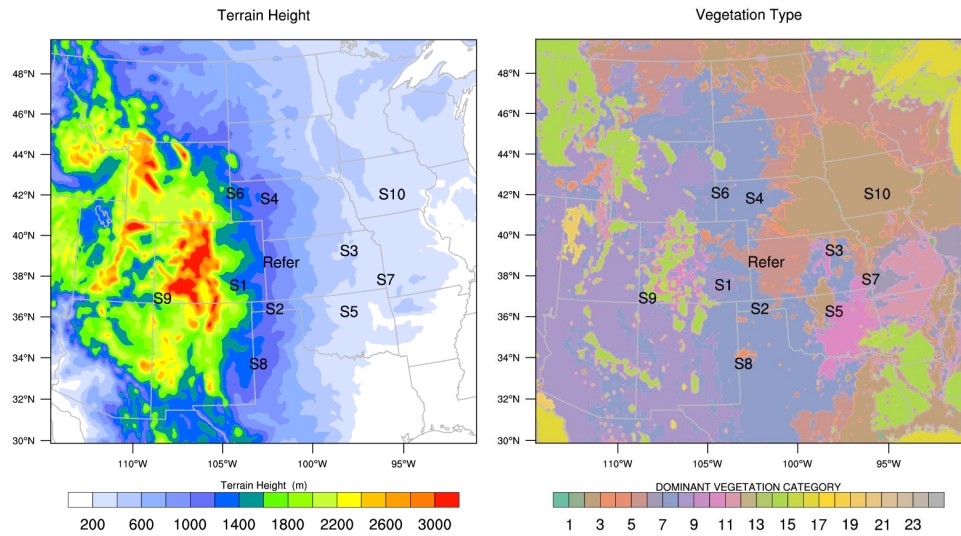


**Figure 8: Terrain height (left) and vegetation types (right) for Logan, Kansas, and other**
**locations that we used to assess the spatial transferability of our domain-aware DNNs.**



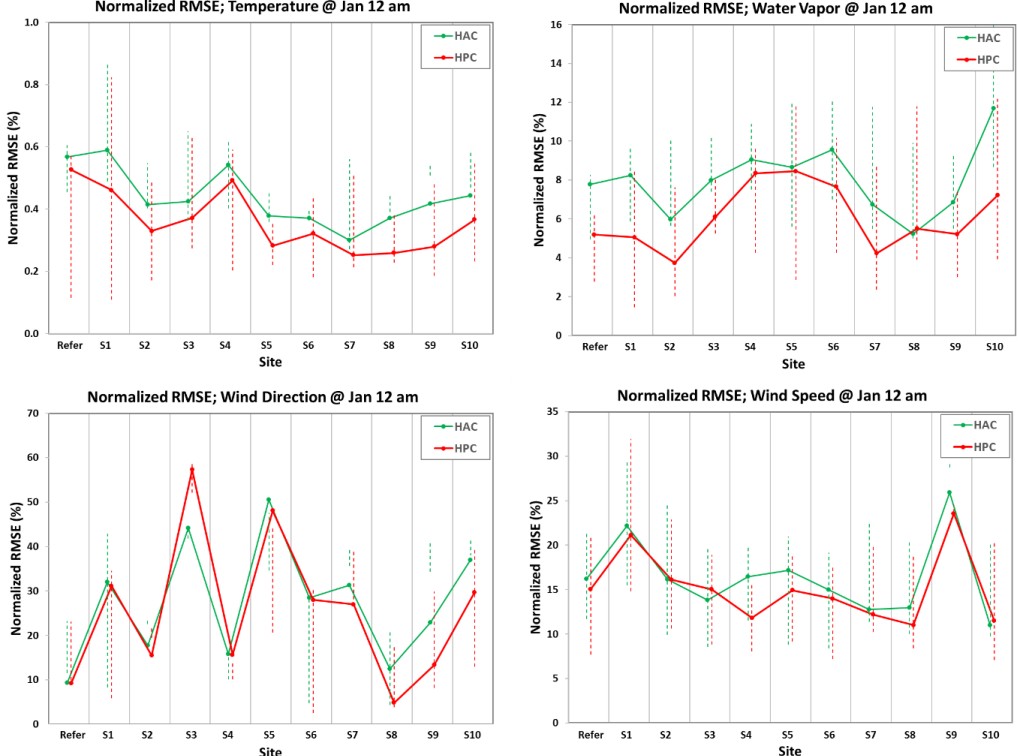


**Figure 9: Normalized RMSEs relative to their corresponding observations at midnight of January for temperature, water vapor mixing ratio, and wind component. The sites are in the order of short to long distance from the reference site at Logan, Kansas.**








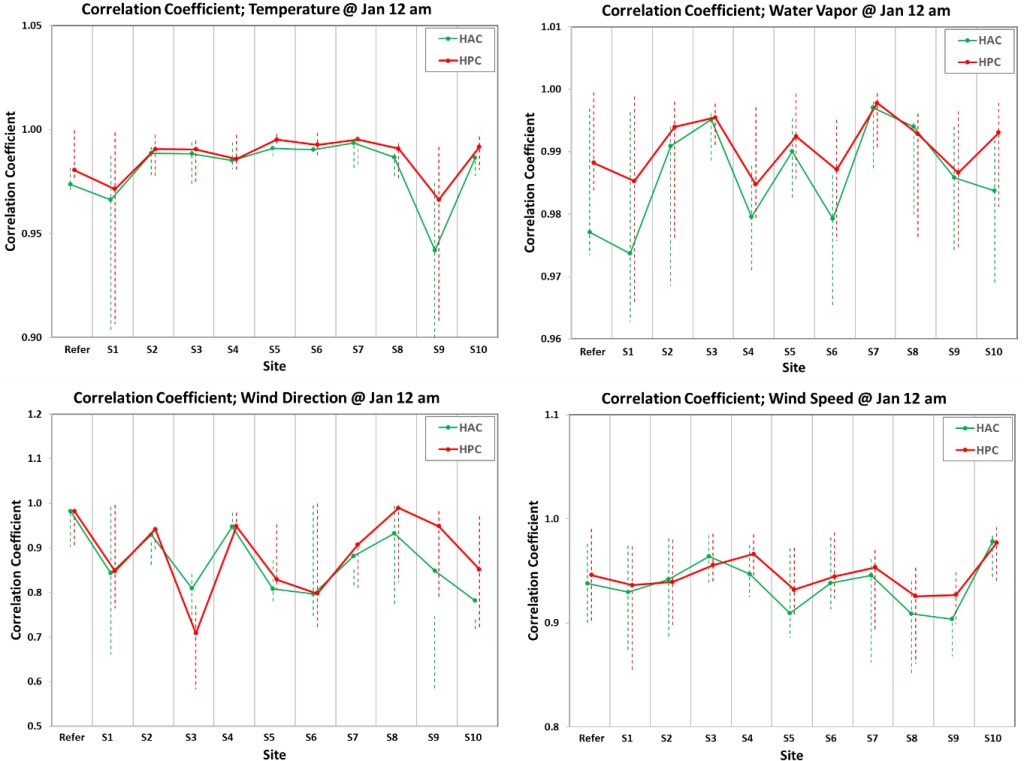


**Figure 10: Same as Figure 9 but for correlations between DNN predictions and observations.**