# Peer review of "boundary layer parameterization in a numerical weather forecast"

_Geoscientific Model Development, 2019_

## Referee Comment (RC1) · Anonymous Referee #1 · 5 Jun 2019

This paper presents a series of neural networks designed to reproduce the output of the YSU PBL parameterization in the WRF model. The goal is to use the NN as a proxy model to reduce the computational cost of running the WRF model. The premise of the work is interesting and worthy of publication. There are several points of clarification and confusion however.

1. The topology of the models is confusing. The FFN is fine, but the size of the hidden layers should be noted. The two hierarchical models appear to be a series of nested single layer neural networks that output at each level. The goal is to enforce layer specificity, though I am not sure why this cannot also happen in the FFN as with what I

assume are a larger number of internal weights and fully connected, it should be able to encode this information as well.

2. I don't understand why the FFN performance is so much worse. If the intermediate layers are sized large enough, then it should have a much larger number of connections and be able to encode more than the hierarchical models. It appears to cut out in training much earlier however. Is this just overfitting due to larger number of connections vs training data? If the amount of data was vastly increased, would we expect FFN to eventually overtake the performance of the hierarchical models?

3. The writeup of the evaluation is a bit confusing. In particular L231-236. I assume it means that you trained using a single grid location, then applied the model to multiple grid points within 800km. If so this should be made more clear. Also why specify individual sites. One could calculate performance on all grid points within 800km. While doing this, it would be useful to see the drop off in performance as a function of distance. The last two plots start down this path, but with the density of points in a model, it should be straight forward to give performance as a function of distance from training point within the 800km range.

4. Can the authors comment on where they see this being put in an online model? It seems like the round trip to and from a GPU (IO) would cause a much bigger delay than just calculating the YSU parameterization in place. Offline this is not as much of a concern as all the data can be preloaded, but when there is a round trip at every model time step, it seems like the IO would become the predominant factor, and not the computation.

5. I assume this is meant to be used as a proxy model for YSU when you are interested in fiddling with a different portion of the model and just want something "good enough" that is computationally cheap (For instance, you're examining microphysical parameterizations, and don't care about PBL explicitly). Is there a concern that the feedback loops with being off by as much as this NN is (up to 60% for some parameters, though much less for most) would cause the output from YSU and this model to diverge quite quickly when being run as a replacement for YSU? If so is this just meant to be a parallel option for a parameterization, or as a drop in for YSU?

––––––––––––––––––––––––––––––

---

## Referee Comment (RC2) · Anonymous Referee #2 · 6 Jun 2019

This is a very interesting work. To the best of my knowledge, the authors are the first researchers who applied machine learning techniques to PBL parameterization. Their results are thought-provoking. Of course, the final evaluation of the developed NN must be performed in parallel runs of WRF with the original PBL parameterization and with the developed NN. However, it is an issue for a separate paper. I believe that this paper should be accepted after some revision and clarifications.

General comments:

1. It is not clear from the text if the authors developed a NN emulation of the PBL parameterization. NN emulation has the same inputs (sometimes augmented by addi-

tional metadata) and the same outputs as the original (in this case YSU) parameterization. Is this the case for the presented study?

2. Domain-aware NN is a confusing term. Which domain are the authors talking about (geographic domain, domain covered by inputs in the input space, etc.)? From the sentence in the paper: "a key drawback of the naïve FFN is that it does not consider the underlying PBL domain structure, such as the patterns that are locality specific and the vertical dependence between different vertical levels of each profile", it can be concluded that it is about vertical correlations between different vertical levels. Probably "domain-aware" name is misplaced (see also comment 5).

3. I cannot completely agree with the aforementioned (in comment 2) sentence from the paper. First, FFN does accounts for vertical correlations in profiles because all level components of profile are built from the same neurons of the previous hidden layer. Second, in this particular study, as is explained in the text, all outputs (including all vertical components of the same profile) are normalized independently, which significantly reduces the sensitivity of FFN (and any NN) to vertical correlations between levels. To preserve vertical dependencies, a profile should be normalized as a whole but not each component independently.

4. The time length of data set to be used for training is not a valuable and universal characteristic. It depends on representativeness of data set, i.e. on how well the variety of atmospheric states is represented in the training set, or how well the domain of input space is sampled. For example, including in the training set more grid points would enrich it with new/different atmospheric states and made more representative. It may shorten the time length of data required for training.

5. Most/all problems with applications to neighboring grid points can be alleviated or completely removed if all grid points (entire grid) are included in training set together with some metadata, i.e. if lat, lon, and the terrain conditions (elevation etc.) are included as additional inputs at each grid point. The NN trained in such a way I'd call

"domain-aware NN".

Specific comments:

1. Is the sizes of the input and output layers are 16 + 85 = 101? (= near-surface variables) and 85 (= 17 vertical levels $\times$ 5 output variables).

2. It is not clear from the text how 17-level profiles produced by NN are integrated with WRF profiles in total 38 level profiles?

3. How many hidden layers have different NNs that the authors use?

4. How many trained parameters (NN weights) has each of these NNs?

5. How many records has the training set that is used?

6. Did not the authors try to train a shallow NN with the same number of weights as the best DNN has?

Without all this information it is difficult to understand why NNs with different architectures perform so differently.

---

## Author Comment (AC1) · 10 Aug 2019

We would like to thank both reviewers for their constructive feedback, which helped us to clarify the presented methods and improve the manuscript significantly.

Please find our response below, and all the changes in the revised manuscript (tracked changes).
presents a series of neural networks designed to reproduce the output of the YSU PBL parameterization in the WRF model. The goal is to use the NN as a proxy model to reduce the computational cost of running the WRF model. The premise of the work is interesting and worthy of publication. There are several points of clarification and confusion however.

1. The topology of the models is confusing. The FFN is fine, but the size of the hidden layers should be noted. The two hierarchical models appear to be a series of nested single layer neural networks that output at each level. The goal is to enforce layer specificity, though I am not sure why this cannot also happen in the FFN as with what I assume are a larger number of internal weights and fully connected, it should be able to encode this information as well.

Response: In the revised manuscript, we detailed the figure description for Figure 1, which shows the architecture of each neural network. We specified that we have 17 hidden layers for all three neural network, indicated by the purple unit in Figure 1. We also revise the manuscript to further emphasize the difference between the three neural networks. The FFN takes all the output data (17 vertical WRF model layer $\times$ 5 variables/layer) and input data for training. The training process doesn't know which data belongs to which layer, the output layer comprises 85 output variables. While this is a typical way to develop neural network, it doesn't consider any vertical mixing in a PBL profile. Therefore, we develop the other two neural networks, which has 17 output layers with each of them having 5 output variables for the particular PBL layer. Each of the 17 hidden layers uses the output from each of these output layers and also the near-surface input, and calculate the output for the next output layer. From the neural network perspective, the key advantage of HPC and HAC over FNN is effective back-propagation while training. In HPC and HAC, each hidden layer has an output layer; therefore, during the back propagation, the gradients from each of the output layer can be used to update the weights of the hidden layer directly to minimize the error for PBL layer specific outputs.

[Figure]

2. I don't understand why the FFN performance is so much worse. If the intermediate layers are sized large enough, then it should have a much larger number of connections and be able to encode more than the hierarchical models. It appears to cut out in training much earlier however. Is this just overfitting due to larger number of connections vs training data? If the amount of data was vastly increased, would we expect FFN to eventually overtake the performance of the hierarchical models?

Response: We clarified this issue and provided an explanation in the revised manuscript. As we mentioned in the response to your first comment, the FFN takes all the output data (17 vertical WRF model layers $\times$ 5 variables per layer) and input data for training. The training process doesn't know which data belongs to which layer, the 85 variables are treated as a whole thing. The key advantage of HPC and HAC over FNN is that, during the training of HPC and HAC, because each of their hidden layer has an output layer, so during the back propagation, the gradients from each of the output layer can be used to update the weights of the hidden layer directly to minimize the error for that particular PBL layer. While for FFN, there is no output layers for each hidden layer, so there is no information that the backward propagation of FFN can take to update weights and minimize errors.

3. The writeup of the evaluation is a bit confusing. In particular L231-236. I assume it means that you trained using a single grid location, then applied the model to multiple grid points within 800km. If so this should be made more clear. Also why specify individual sites. One could calculate performance on all grid points within 800km. While doing this, it would be useful to see the drop off in performance as a function of distance. The last two plots start down this path, but with the density of points in a model, it should be straight forward to give performance as a function of distance from training point within the 800km range.

Response: Thanks for your suggestions. We have modified the text in line 231-236 to clarify that we trained our DNNs at a single location (e.g, Logan, Kansas) and then we apply the DNN to multiple grid points nearby. We also update the last two figures and the corresponding discussions. Instead of picking several stations, we test the DNN models over all the grid points of an 1100 × 1100 km area, with 90×91 grid points. To reduce the computing burden, we pick every other 7 grid points and get 13×13 grid points over the area, which still maintain the terrain height variability. Then we calculate RMSE and Pearson correlations of the neural network predictions compared with observations (here WRF model simulation), and see how RMSE and correlation change with distance from the center (where we develop the DNNs) of the 13x13 area. Results show that the neural network can be used by other locations if they are not far, and they have very similar terrain conditions for temperature, water vapor and wind speed. For wind direction, to use the same neural network, the nearby grid points should also under the same weather regimes, such as the large-scale circulations, etc. This indicates that, it's not always safe to train one model over a region unless the region has homogenous features in every regard we discussed here.

If we trained the network over a larger domain, we may need larger dataset for training. There is likely an optimum domain size over which a network would be useful. Therefore, we think that there may be a small number of region specific networks necessary for representing the PBL process for our entire model coverage of North American continent. The transferability tests described in the paper are one approach that we can use to determine the size of a single DNN model and its application extent. Developing DNNs for the whole region will require significantly larger computing resources. These ideas are beyond the scope of this study and will be explored in the future.

4. Can the authors comment on where they see this being put in an online model? It seems like the round trip to and from a GPU (IO) would cause a much bigger delay than just calculating the YSU parameterization in place. Offline this is not as much of a concern as all the data can be preloaded, but when there is a round trip at every model time step, it seems like the IO would become the predominant factor, and not the computation.

Response: We need GPUs for faster training. In a deployment scenario, as shown in several deep learning case studies, CPUs are enough for fast inference/prediction. In that case, we do not need GPUs and can avoid the data movement cost. Moreover, we are anticipating that the next HPC platforms that will be available will be 'accelerated' CPUs (e.g. the exascale systems at the DOE leadership computing facilities at Argonne National Laboratory and Oak Ridge Laboratory). The goal of the accelerated CPU architecture is to decrease the IO costs and make the GPU an integral part of the CPU design. The types of models we discuss here will be highly suitable for these machines.

5. I assume this is meant to be used as a proxy model for YSU when you are interested in fiddling with a different portion of the model and just want something "good enough" that is computationally cheap (For instance, you're examining microphysical parameterizations, and don't care about PBL explicitly). Is there a concern that the feedback loops with being off by as much as this NN is (up to 60% for some parameters, though much less for most) would cause the output from YSU and this model to diverge quite quickly when being run as a replacement for YSU? If so is this just meant to be a parallel option for a parameterization, or as a drop in for YSU?

Response: The expectation we have is that these types of DNN models could function as a drop-in replacement for existing parameterizations. We have trained the model with a limited amount of grid cells as a proof of concept. Eventually, this model will be trained for all regions and conditions for the extensive simulation database we have. The goal is to develop DNN emulators for all the expensive parts of the model (radiation, microphysics, cumulus etc.) that would function as a high-spatial resolution 'emulator' of the model. Another possible path is to develop an emulator for the entire model disregarding each process (e.g. Scher, JGR 2019). We expect both paths to lead to the development of emulators that will be critical for generating larger ensemble of model simulations for uncertainty quantification in future climate projections.

Reference:

[Figure]

Scher, S.: Toward data-driven weather and climate forecasting: Approximating a simple general circulation model with deep learning. Geophysical Research Letters, 45, 12,616–12,622, 2018.

Anonymous Referee #2

This is a very interesting work. To the best of my knowledge, the authors are the first researchers who applied machine learning techniques to PBL parameterization. Their results are thought-provoking. Of course, the final evaluation of the developed NN must be performed in parallel runs of WRF with the original PBL parameterization and with the developed NN. However, it is an issue for a separate paper. I believe that this paper should be accepted after some revision and clarifications.

General comments:

1. It is not clear from the text if the authors developed a NN emulation of the PBL parameterization. NN emulation has the same inputs (sometimes augmented by additional metadata) and the same outputs as the original (in this case YSU) parameterization. Is this the case for the presented study?

Response: The DNN developed here is an emulator in the sense that it is trained using the output from YSU scheme (not MYJ or MYNN PBL scheme). The inputs for our DNNs closely correspond to variables that are used as inputs to the YSU scheme in the WRF model.

2. Domain-aware NN is a confusing term. Which domain are the authors talking about (geographic domain, domain covered by inputs in the input space, etc.)? From the sentence in the paper: "a key drawback of the naïve FFN is that it does not consider the underlying PBL domain structure, such as the patterns that are locality specific and the vertical dependence between different vertical levels of each profile", it can be concluded that it is about vertical correlations between different vertical levels. Probably

"domain-aware" name is misplaced (see also comment 5).

Response: We have used the word "domain-aware" to mean subject expertise and the word "domain" to mean the domain of science rather than a geographical or spatial domain. Thus, one of the goals of this study is to show the importance of collaborations between data scientists and domain science experts. We first develop a neural network without any additional insights from a domain expert, such as local and nonlocal mixing in the vertical direction but purely driven by a knowledge of the key inputs to the YSU scheme. We then develop neural network that incorporate domain expertise, and we consider both the local and non-local mixing by taking into account the connection between one certain layer and the previous one layer (HPC) and the previous all layers (HAC) as well as the near-surface variable as inputs. This leads to a significant improvement of the prediction accuracy. We had an explanation of the use of domain-aware in the abstract; in the revised manuscript; we made an effort to further explain it by pointing the nonlocal mixing, which are vital for the PBL process to capture the turbulence in the lower troposphere.

As we respond to Reviewer #1, the FFN takes all the output data (17 vertical WRF model layer x 5 variables/layer) and input data for training. The training process doesn't know which data (among the 85 variables) belongs to which layer, the output layer of FFN consider all 85 variables as a whole thing. While this is a typical way to develop neural network, it doesn't consider any vertical mixing in a PBL layer. Therefore, we develop the other two neural networks, which have 17 output layers with each of them having 5 variables for the particular PBL layer. Each hidden layer uses the output from each of these output layers and also the near-surface input, and calculate the output for the next output layer. From the neural network perspective, the key advantage of HPC and HAC over FNN is effective back-propagation while training. In HPC and HAC, each hidden layer has an output layer; so during the back propagation, the gradients from each of the output layer can be used to update the weights of the hidden layer directly to minimize the error for PBL layer specific outputs.

In summary, we still keep the domain-aware term, but we clarified that the domain-aware is about considering local and non-local mixing in the PBL.

3. I cannot completely agree with the aforementioned (in comment 2) sentence from the paper. First, FFN does accounts for vertical correlations in profiles because all level components of profile are built from the same neurons of the previous hidden layer. Second, in this particular study, as is explained in the text, all outputs (including all vertical components of the same profile) are normalized independently, which significantly reduces the sensitivity of FFN (and any NN) to vertical correlations between levels. To preserve vertical dependencies, a profile should be normalized as a whole but not each component independently.

Response: In the revised manuscript, we made an efforts to clarify the difference between FFN and HAC/HPC in both text and Figure 1. As we response to your comment 2, FFN takes all the output data (17 vertical WRF model layer x 5 variables/layer) and input data for training. The training process doesn't know which data belongs to which layer, the 85 output variables are treated as a whole. So it doesn't consider any vertical correlations in the profile.

We apologize for any unclear text in the original manuscript. we normalize each output variable independently, not each vertical layer independently. In other words, we normalize the whole profile of each variable separately, because the values of the five output variables are in different scale (range of values). The normalization is done per output variable so that they all have the same scale. This is a common approach in NN training as it allows the back propagation to treat the errors equally. Note that for prediction, we apply inverse transformation and compute the prediction error ($R^2$ and RMSE) in the original scale.

4. The time length of data set to be used for training is not a valuable and universal characteristic. It depends on representativeness of data set, i.e. on how well the variety of atmospheric states is represented in the training set, or how well the domain of input space is sampled. For example, including in the training set more grid points would enrich it with new/different atmospheric states and made more representative. It may shorten the time length of data required for training.

Response: Thanks for your comment. Reviewer #1 also had a similar insight. We do agree that the neural network developed in this study based on the individual location (Logan, Kansas) likely will not be applicable universally. However, this model will have a region of applicability that can be tested using similar approach to those discussed in the manuscript. We may need several such networks to cover the entire model simulated region or train the DNN with data from the entire model simulated region. The later option as explained will need to performed on HPC systems and will be the target of our future research.

We agree that training more grid points (in space) would enrich the dataset, but it also have the risk of introducing more noise for the neural network, unless the regions is homogenous. This might be done over a very small region by taking several grid points but not a relatively large region. By homogenous we mean that different grid points should also under the same weather regimes, such as the large scale circulations, etc. The reason we say this is that, from our spatial transferability analysis, we found the neural network can be used for temperature, water vapor, and even wind speed over other locations as far as 500km, but for wind direction, the different grid points should also under the same circulation patterns (for example, if the wind is driven by terrain over one location, then the network doesn't apply to locations that are driven by large-scale circulations). On the other hand, we do see it is worthwhile to develop DNNs for the whole region instead of individual locations. As we mentioned in our discussion, however, this will require additional computational consideration on HPC and will be considered in the follow-on effort.

5. Most/all problems with applications to neighboring grid points can be alleviated or completely removed if all grid points (entire grid) are included in training set together with some metadata, i.e. if lat, lon, and the terrain conditions (elevation etc.) are
included as additional inputs at each grid point. The NN trained in such a way I'd call "domain-aware NN".

Response: Thanks for the suggestion and we found it's very helpful. In our revised manuscript, for the last two figures, instead of picking several stations we test the DNN models over all the grids of a 1100x1100 km area, with 90x91 grid points over that area. To reduce the computing burden, we pick every other 7 grid points and get 13x13 grid points over the area, which still maintain the terrain height variability. Then we calculate RMSE and correlations of the neural network prediction compared with observations (here WRF model simulation), and see how RMSE and correlation change with distance from the center (where we develop the DNNs) of the 13x13 area.

As we respond to your comment 2, we use the word 'domain' in the sense of domain-science expertise and not spatial domain. In other words, our neural networks were not developed considering spatial domain factors, they were developed only based on individual locations. It is referred to domain knowledges about the PBL structure, specifically, the local and nonlocal mixing of turbulence in the lower troposphere.

Specific comments:

1. Is the sizes of the input and output layers are 16 + 85 = 101? (= near-surface variables) and 85 (= 17 vertical levels _ 5 output variables).

Response: For FFN, we have an input layer, which has 16 near-surface variables; we have 17 hidden layers; and one output layer, which has 85 variables (5 variables for each of the 17 WRF vertical layer).

For HPC, we have 16 near-surface variables as one part of the input, and we also use the output (5 variables) of each previous hidden layer as input for the next hidden layer. We have 17 hidden layers, and 17 output layers. HAC is similar to HPC, but uses the output of ALL the previous hidden layer as input for the next hidden layer. We have specified this in text and also added clarifications in the caption of Figure 1.

[Figure]

2. It is not clear from the text how 17-level profiles produced by NN are integrated with WRF profiles in total 38 level profiles?

Response: The middle and upper troposphere (all layers above the PBL) are considered fully resolved by the dynamics simulated by the WRF model. So the upper 21 layers will be still from the WRF model itself. There may be discontinuity between the 17th and the 18th layer (which are from NN and the WRF, respectively), and need to be smoothed. This will be future study when we implement the NNs into WRF.

3. How many hidden layers have different NNs that the authors use?

Response: we used 17 hidden layers for all three NNs developed in this study. We add this information in the revised manuscript. The hidden layers are represented by the purple unit in Figure 1.

4. How many trained parameters (NN weights) has each of these NNs?

Response: for FNN we have 10,693 trained parameters; for HPC we have 16,597 trained parameters, and for HAC we have 26,197 trained parameters.

5. How many records has the training set that is used?

Response: We have described this in data at line 120-125 in the original manuscript. in the revised manuscript we move this description to 2.3 Setup as following: "The 22-year data from the WRF simulation was partitioned into three parts: a training set consisting of 20 years (1984–2003) of 3-hourly data to train the NN; a development set (also called validation set) consisting of 1 year (2004) of 3-hourly data used to tune the algorithm's hyperparameters and to control overfitting (the situation where the trained network predicts well on the training data but not on the test data); and a test set consisting of 1 year of records (2005) for prediction and evaluations."

6. Did not the authors try to train a shallow NN with the same number of weights as the best DNN has? Without all this information it is difficult to understand why NNs with different architectures perform so differently.
[Figure]

Response: as we explained earlier, the FFN takes all the output data (17 vertical WRF model layer x 5 variables/layer) and input data for training. The training process doesn't know which data belongs to which layer, the 85 variables are treated as a whole. While this is a typical way to develop neural network, it doesn't consider any vertical mixing in a PBL layer. Therefore, we develop the other two neural networks, which has 17 output layers with each of them having 5 variables for the particular PBL layer. Each hidden layer uses the output from each of these output layers and also the near-surface input, and calculate the output for the next output layer. From the neural network perspective, the key advantage of HPC and HAC over FNN is effective back-propagation while training. In HPC and HAC, each hidden layer has an output layer; consequently, during the back propagation, the gradients from each of the output layer can be used to update the weights of the hidden layer directly to minimize the error for PBL layer specific outputs.

Please also note the supplement to this comment:
https://www.geosci-model-dev-discuss.net/gmd-2019-79/gmd-2019-79-AC1-supplement.pdf

**Supplement:**

This paper presents a series of neural networks designed to reproduce the output of the YSU PBL parameterization in the WRF model. The goal is to use the NN as a proxy model to reduce the computational cost of running the WRF model. The premise of the work is interesting and worthy of publication. There are several points of clarification and confusion however.

1. The topology of the models is confusing. The FFN is fine, but the size of the hidden layers should be noted. The two hierarchical models appear to be a series of nested single layer neural networks that output at each level. The goal is to enforce layer specificity, though I am not sure why this cannot also happen in the FFN as with what I assume are a larger number of internal weights and fully connected, it should be able to encode this information as well.
**Response:** In the revised manuscript, we detailed the figure description for Figure 1, which shows the architecture of each neural network. We specified that we have 17 hidden layers for all three neural network, indicated by the purple unit in Figure 1. We also revise the manuscript to further emphasize the difference between the three neural networks. The FFN takes all the output data (17 vertical WRF model layer × 5 variables/layer) and input data for training. The training process doesn't know which data belongs to which layer, the output layer comprises 85 output variables. While this is a typical way to develop neural network, it doesn't consider any vertical mixing in a PBL profile. Therefore, we develop the other two neural networks, which has 17 output layers with each of them having 5 output variables for the particular PBL layer. Each of the 17 hidden layers uses the output from each of these output layers and also the near-surface input, and calculate the output for the next output layer. From the neural network perspective, the key advantage of HPC and HAC over FNN is effective back-propagation while training. In HPC and HAC, each hidden layer has an output layer; therefore, during the back propagation, the gradients from each of the output layer can be used to update the weights of the hidden layer directly to minimize the error for PBL layer specific outputs.

2. I don't understand why the FFN performance is so much worse. If the intermediate layers are sized large enough, then it should have a much larger number of connections and be able to encode more than the hierarchical models. It appears to cut out in training much earlier however. Is this just overfitting due to larger number of connections vs training data? If the amount of data was vastly increased, would we expect FFN to eventually overtake the performance of the hierarchical models?
**Response:** We clarified this issue and provided an explanation in the revised manuscript. As we mentioned in the response to your first comment, the FFN takes *all* the output data (17 vertical WRF model layers × 5 variables per layer) and input data for training. The training process doesn't know which data belongs to which layer, the 85 variables are treated as a *whole thing*. The key advantage of HPC and HAC over FNN is that, during the training of HPC and HAC, because each of their hidden layer has an output layer, so during the back propagation, the gradients from each of the output layer can be used to update the weights of the hidden layer directly to minimize the error for that particular PBL layer. While for FFN, there is no output layers for each hidden layer, so there is no information that the backward propagation of FFN can take to update weights and minimize errors.

3. The writeup of the evaluation is a bit confusing. In particular L231-236. I assume it means that you trained using a single grid location, then applied the model to multiple grid points within 800km. If so this should be made more clear. Also why specify individual sites. One could calculate performance on all grid points within 800km. While doing this, it would be useful to see the drop off in performance as a function of distance. The last two plots start down this path, but with the density of points in a model, it should be straight forward to give performance as a function of distance from training point within the 800km range.

**Response:** Thanks for your suggestions. We have modified the text in line 231-236 to clarify that we trained our DNNs at a single location (e.g, Logan, Kansas) and then we apply the DNN to multiple grid points nearby. We also update the last two figures and the corresponding discussions. Instead of picking several stations, we test the DNN models over all the grid points of an $1100 \times 1100$ km area, with $90 \times 91$ grid points. To reduce the computing burden, we pick every other 7 grid points and get $13 \times 13$ grid points over the area, which still maintain the terrain height variability. Then we calculate RMSE and Pearson correlations of the neural network predictions compared with observations (here WRF model simulation), and see how RMSE and correlation change with distance from the center (where we develop the DNNs) of the 13x13 area. Results show that the neural network can be used by other locations if they are not far, and they have very similar terrain conditions for temperature, water vapor and wind speed. For wind direction, to use the same neural network, the nearby grid points should also under the same weather regimes, such as the large-scale circulations, etc. This indicates that, it's not always safe to train one model over a region unless the region has homogenous features in every regard we discussed here.

If we trained the network over a larger domain, we may need larger dataset for training. There is likely an optimum domain size over which a network would be useful. Therefore, we think that there may be a small number of region specific networks necessary for representing the PBL process for our entire model coverage of North American continent. The transferability tests described in the paper are one approach that we can use to determine the size of a single DNN model and its application extent. Developing DNNs for the whole region will require significantly larger computing resources. These ideas are beyond the scope of this study and will be explored in the future.

4. Can the authors comment on where they see this being put in an online model? It seems like the round trip to and from a GPU (IO) would cause a much bigger delay than just calculating the YSU parameterization in place. Offline this is not as much of a concern as all the data can be preloaded, but when there is a round trip at every model time step, it seems like the IO would become the predominant factor, and not the computation.

**Response:** We need GPUs for faster training. In a deployment scenario, as shown in several deep learning case studies, CPUs are enough for fast inference/prediction. In that case, we do not need GPUs and can avoid the data movement cost. Moreover, we are anticipating that the next HPC platforms that will be available will be 'accelerated' CPUs (e.g. the exascale systems at the DOE leadership computing facilities at Argonne National Laboratory and Oak Ridge Laboratory). The goal of the accelerated CPU architecture is to decrease the IO costs and make the GPU an integral part of the CPU design. The types of models we discuss here will be highly suitable for these machines.

5. I assume this is meant to be used as a proxy model for YSU when you are interested in fiddling with a different portion of the model and just want something "good enough" that is computationally cheap (For instance, you're examining microphysical parameterizations, and don't care about PBL explicitly). Is there a concern that the feedback loops with being off by as much as this NN is (up to 60% for some parameters, though much less for most) would cause the output from YSU and this model to diverge quite quickly when being run as a replacement for YSU? If so is this just meant to be a parallel option for a parameterization, or as a drop in for YSU?

**Response:** The expectation we have is that these types of DNN models could function as a drop-in replacement for existing parameterizations. We have trained the model with a limited amount of grid cells as a proof of concept. Eventually, this model will be trained for all regions and conditions for the extensive simulation database we have. The goal is to develop DNN emulators for all the expensive parts of the model (radiation, microphysics, cumulus etc.) that would function as a high-spatial resolution 'emulator' of the model. Another possible path is to develop an emulator for the entire model disregarding each process (e.g. Scher, JGR 2019). We expect both paths to lead to the development of emulators that will be critical for generating larger ensemble of model simulations for uncertainty quantification in future climate projections.

**Response**: We have used the word "domain-aware" to mean subject expertise and the word "domain" to mean the domain of science rather than a geographical or spatial domain. Thus, one of the goals of this study is to show the importance of collaborations between data scientists and domain science experts. We first develop a neural network without any additional insights from a domain expert, such as local and nonlocal mixing in the vertical direction but purely driven by a knowledge of the key inputs to the YSU scheme. We then develop neural network that incorporate domain expertise, and we consider both the local and non-local mixing by taking into account the connection between one certain layer and the previous one layer (HPC) and the previous all layers (HAC) as well as the near-surface variable as inputs. This leads to a significant improvement of the prediction accuracy. We had an explanation of the use of domain-aware in the abstract; in the revised manuscript; we made an effort to further explain it by pointing the nonlocal mixing, which are vital for the PBL process to capture the turbulence in the lower troposphere.

As we respond to Reviewer #1, the FFN takes all the output data (17 vertical WRF model layer x 5 variables/layer) and input data for training. The training process doesn't know which data (among the 85 variables) belongs to which layer, the output layer of FFN consider all 85 variables as a whole thing. While this is a typical way to develop neural network, it doesn't consider any vertical mixing in a PBL layer. Therefore, we develop the other two neural networks, which have 17 output layers with each of them having 5 variables for the particular PBL layer. Each hidden layer uses the output from each of these output layers and also the near-surface input, and calculate the output for the next output layer. From the neural network perspective, the key advantage of HPC and HAC over FNN is effective back-propagation while training. In HPC and HAC, each hidden layer has an output layer; so during the back propagation, the gradients from each of the output layer can be used to update the weights of the hidden layer directly to minimize the error for PBL layer specific outputs.

In summary, we still keep the domain-aware term, but we clarified that the domain-aware is about considering local and non-local mixing in the PBL.

3. I cannot completely agree with the aforementioned (in comment 2) sentence from the paper. First, FFN does accounts for vertical correlations in profiles because all level components of profile are built from the same neurons of the previous hidden layer. Second, in this particular study, as is explained in the text, all outputs (including all vertical components of the same profile) are normalized independently, which significantly reduces the sensitivity of FFN (and any NN) to vertical correlations between levels. To preserve vertical dependencies, a profile should be normalized as a whole but not each component independently.

**Response:** In the revised manuscript, we made an efforts to clarify the difference between FFN and HAC/HPC in both text and Figure 1. As we response to your comment 2, FFN takes all the output data (17 vertical WRF model layer x 5 variables/layer) and input data for training. The training process doesn't know which data belongs to which layer, the 85 output variables are treated as a whole. So it doesn't consider any vertical correlations in the profile.

We apologize for any unclear text in the original manuscript. we normalize each output variable independently, not each vertical layer independently. In other words, we normalize the whole profile of each variable separately, because the values of the five output variables are in different scale (range of values). The normalization is done per output variable so that they all have the same scale. This is a common approach in NN training as it allows the back propagation to treat the errors equally. Note that for prediction, we apply inverse transformation and compute the prediction error ($R^2$ and RMSE) in the original scale.

4. The time length of data set to be used for training is not a valuable and universal characteristic. It depends on representativeness of data set, i.e. on how well the variety of atmospheric states is represented in the training set, or how well the domain of input space is sampled. For example, including in the training set more grid points would enrich it with new/different atmospheric states and made more representative. It may shorten the time length of data required for training.

**Response**: Thanks for your comment. Reviewer #1 also had a similar insight. We do agree that the neural network developed in this study based on the individual location (Logan, Kansas) likely will not be applicable universally. However, this model will have a region of applicability that can be tested using similar approach to those discussed in the manuscript. We may need several such networks to cover the entire model simulated region or train the DNN with data from the entire model simulated region. The later option as explained will need to performed on HPC systems and will be the target of our future research.

We agree that training more grid points (in space) would enrich the dataset, but it also have the risk of introducing more noise for the neural network, unless the regions is homogenous. This might be done over a very small region by taking several grid points but not a relatively large region. By homogenous we mean that different grid points should also under the same weather regimes, such as the large scale circulations, etc. The reason we say this is that, from our spatial transferability analysis, we found the neural network can be used for temperature, water vapor, and even wind speed over other locations as far as 500km, but for wind direction, the different grid points should also under the same circulation patterns (for example, if the wind is driven by terrain over one location, then the network doesn't apply to locations that are driven by large-scale circulations). On the other hand, we do see it is worthwhile to develop DNNs for the whole region instead of individual locations. As we mentioned in our discussion, however, this will require additional computational consideration on HPC and will be considered in the follow-on effort.

5. Most/all problems with applications to neighboring grid points can be alleviated or completely removed if all grid points (entire grid) are included in training set together with some metadata, i.e. if lat, lon, and the terrain conditions (elevation etc.) are included as additional inputs at each grid point. The NN trained in such a way I'd call "domain-aware NN".

**Response**: Thanks for the suggestion and we found it's very helpful. In our revised manuscript, for the last two figures, instead of picking several stations we test the DNN models over all the grids of a 1100x1100 km area, with 90x91 grid points over that area. To reduce the computing burden, we pick every other 7 grid points and get 13x13 grid points over the area, which still maintain the terrain height variability. Then we calculate RMSE and correlations of the neural network prediction compared with observations (here WRF model simulation), and see how RMSE and correlation change with distance from the center (where we develop the DNNs) of the 13x13 area.

As we respond to your comment 2, we use the word 'domain' in the sense of domain-science expertise and not spatial domain. In other words, our neural networks were not developed considering spatial domain factors, they were developed only based on individual locations. It is referred to domain knowledges about the PBL structure, specifically, the local and nonlocal mixing of turbulence in the lower troposphere.

Specific comments:
1. Is the sizes of the input and output layers are 16 + 85 = 101? (= near-surface variables) and 85 (= 17 vertical levels _ 5 output variables).

**Response**: For FFN, we have an input layer, which has 16 near-surface variables; we have 17 hidden layers; and one output layer, which has 85 variables (5 variables for each of the 17 WRF vertical layer).

For HPC, we have 16 near-surface variables as one part of the input, and we also use the output (5 variables) of each previous hidden layer as input for the next hidden layer. We have 17 hidden layers, and 17 output layers. HAC is similar to HPC, but uses the output of ALL the previous hidden layer as input for the next hidden layer.

We have specified this in text and also added clarifications in the caption of Figure 1.

2. It is not clear from the text how 17-level profiles produced by NN are integrated with WRF profiles in total 38 level profiles?

**Response:** The middle and upper troposphere (all layers above the PBL) are considered fully resolved by the dynamics simulated by the WRF model. So the upper 21 layers will be still from the WRF model itself. There may be discontinuity between the 17th and the 18th layer (which are from NN and the WRF, respectively), and need to be smoothed. This will be future study when we implement the NNs into WRF.

3. How many hidden layers have different NNs that the authors use?

**Response**: we used 17 hidden layers for all three NNs developed in this study. We add this information in the revised manuscript. The hidden layers are represented by the purple unit in Figure 1.

4. How many trained parameters (NN weights) has each of these NNs?

**Response**: for FNN we have 10,693 trained parameters; for HPC we have 16,597 trained parameters, and for HAC we have 26,197 trained parameters.

5. How many records has the training set that is used?

**Response:** We have described this in data at line 120-125 in the original manuscript. in the revised manuscript we move this description to **2.3 Setup** as following:

"The 22-year data from the WRF simulation was partitioned into three parts: a training set consisting of 20 years (1984–2003) of 3-hourly data to train the NN; a development set (also called validation set) consisting of 1 year (2004) of 3-hourly data used to tune the algorithm's hyperparameters and to control overfitting (the situation where the trained network predicts well on the training data but not on the test data); and a test set consisting of 1 year of records (2005) for prediction and evaluations."

6. Did not the authors try to train a shallow NN with the same number of weights as the best DNN has? Without all this information it is difficult to understand why NNs with different architectures perform so differently.

**Response:** as we explained earlier, the FFN takes all the output data (17 vertical WRF model layer x 5 variables/layer) and input data for training. The training process doesn't know which data belongs to which layer, the 85 variables are treated as a whole. While this is a typical way to develop neural network, it doesn't consider any vertical mixing in a PBL layer. Therefore, we develop the other two neural networks, which has 17 output layers with each of them having 5 variables for the particular PBL layer. Each hidden layer uses the output from each of these output layers and also the near-surface input, and calculate the output for the next output layer. 
[revised manuscript text omitted]